# Gaseous elemental mercury (GEM) fluxes over canopy of two typical subtropical forests in south China

Qian Yu[1], Yao Luo[1], Shuxiao Wang[1,2], Zhiqi Wang[1], Jiming Hao[1,2], Lei Duan[1,2]

[1]State Key Laboratory of Environmental Simulation and Pollution Control, School of Environment, Tsinghua University, Beijing 100084, China.

[2]Collaborative Innovation Centre for Regional Environmental Quality, Tsinghua University, Beijing 100084, China.

*Correspondence to*: Lei Duan (lduan@tsinghua.edu.cn)

**Abstract.** Mercury (Hg) exchange between forests and the atmosphere plays an important role in global Hg cycling. The present estimate of global emission of Hg from natural source has large uncertainty partly due to the lack of chronical and valid field data, particularly for terrestrial surfaces in China, the most important contributor to global atmospheric Hg. In this study, micrometeorological method (MM) was used to continuously observe gaseous elemental mercury (GEM) fluxes over forest canopy at a mildly polluted site (Qianyanzhou, QYZ) and a moderately polluted site (Huitong, HT, near a large Hg mine) in subtropical south China for a full year from January to December in 2014. The GEM flux measurements over forest canopy in QYZ and HT showed net emission with annual average values of 6.67 and 0.30 ng m$^{-2}$ h$^{-1}$ respectively. Daily variations of GEM fluxes showed an increasing emission with the increasing air temperature and solar radiation in the daytime to a peak at 1:00 pm, and decreasing emission thereafter, even as a GEM sink or balance at night. High temperature and low air Hg concentration resulted in the high Hg emission in summer. Low temperature in winter and Hg absorption by plant in spring resulted in low Hg emission, or even adsorption in the two seasons. GEM fluxes were positively correlated with air temperature, soil temperature, wind speed, and solar radiation while negatively correlated with air humidity and atmospheric GEM concentration. The lower emission fluxes of GEM at the moderately polluted site (HT) when comparing with that in the mildly polluted site (QYZ), may result from a much higher adsorption fluxes at night in spite of a similar or higher emission fluxes during daytime. It testified that the higher atmospheric GEM concentration at HT restricted the forest GEM emission. Great attention should be paid on forest as a critical increasing Hg emission source with the decreasing atmospheric GEM concentration in polluted area because of the Hg emission abatement in the future.

## 1 Introduction

Mercury (Hg) is a world-wide concerned environmental contaminant due to its cyclic transport between air, water, soil, and the biosphere, and its tendency to bioaccumulate in the environment as neurotoxic mono-methylated compounds(CH$_3$Hg-) (Driscoll et al., 2013), which can cause damage to the environment and human health (Lindqvist et al., 1991). Atmospheric Hg exists in three different forms with different chemical and physical properties: gaseous elemental mercury (GEM, Hg$^0$), gaseous oxidized mercury (GOM, Hg$^{2+}$), and particulate-bound mercury (PBM, Hg$^p$). Because of its mild reactivity, high

volatility, and low dry deposition velocity and water solubility, GEM is the most abundant form of Hg in the atmosphere
(Gustin and Jaffe, 2010; Holmes et al., 2010), and can long-distance transport due to the long residence time (0.5~2 yr)
(Schroeder et al., 1998).
Hg emission flux from anthropogenic sources has been quantified with reasonable consistency from 1900 to 2500 t yr-1 (Streets
et al., 2009; Streets et al., 2011; Zhang et al., 2015; Zhang et al., 2016). However, the present estimates of natural Hg emission
from waters, soils, and vegetation are poorly constrained and have large uncertainties, with the values larger than anthropogenic
emission (e.g., 2000 t yr$^{-1}$, Lindqvist et al., 1991; 5207 t yr$^{-1}$, Pirrone et al., 2010; 4080~6950 t yr$^{-1}$, UNEP, 2013; 4380~6630
t yr$^{-1}$ Zhu et al., 2016). The reliable quantification of natural Hg source, specifically GEM exchange between terrestrial
ecosystem and the atmosphere would contribute to the understanding of global and regional Hg cycling budgets (Pirrone et al.,
2010; Wang et al., 2014b; Song et al., 2015).
As a dominant ecosystem on the Earth, forest is generally regarded as an active pool of Hg (Lindberg et al., 2007; Ericksen et
al., 2003; Sigler et al., 2009). Hg in the forest ecosystem is derived primarily from atmospheric deposition (Grigal, 2003), and
foliar uptake of GEM has been recognized as a principal pathway for atmospheric Hg to enter terrestrial ecosystems (Frescholtz
et al., 2003; Niu et al., 2011; Obrist, 2007). Accumulated Hg in foliage is transferred to soil reservoirs via plant detritus (St
Louis et al., 2001) or may partially be released back into the atmosphere (Bash and Miller, 2009). In addition, Hg may enter
the foliage by recycling processes, releasing GEM from underlying soil surfaces (Millhollen et al., 2006b). Soil–air GEM
exchange is controlled by numerous factors including physicochemical properties of soil substrate and abiotic/biotic processes
in the soil, meteorological conditions, and atmospheric composition (Bahlmann et al., 2006; Carpi and Lindberg, 1997; Engle
et al., 2005; Fritsche et al., 2008a; Gustin, 2011; Rinklebe et al., 2010; Mauclair et al., 2008; Zhang et al., 2008). The majority
of reported GEM flux measurements over terrestrial soils indicated net emission in warmer seasons and near-zero fluxes at
cold temperatures (Sommar et al., 2013). There are ongoing debates regarding whether or not forest is a sink or a source of
GEM because the forest/air exchange flux is the sum of vegetation and soil exchange flux, depending on not only atmospheric
concentration and meteorological conditions, but also plant community composition (Bash and Miller, 2009; Converse et al.,
2010) over shorter or longer periods.
China is currently the world's top emitter of anthropogenic Hg with a value of 538t in 2010 (Zhang et al., 2015) and 530t in
2014 (Wu et al., 2016), which resulted in an elevated Hg deposition to terrestrial ecosystem and thus Hg accumulate in land
surface. Given the forest is likely to have large GEM re-emission of legacy Hg stored through old-deposition, it is important
to know the role of forests in China in global Hg transport and cycle.  However, there are far fewer long-time studies of forest
GEM exchange flux in China, especially for the subtropical forest, which is unique in the world. In this study, directly
measurements of net exchange of GEM over canopy of subtropical forests was conducted at a relatively mildly polluted site
and a moderately polluted site impacted by an adjacent Hg mine in south China. The objective of this study is to quantify the
natural Hg emission from the typical forest ecosystems, and analyse its influencing factors.

## 2 Materials and methods

### 2.1 Site description

This study was conducted at Qianyanzhou (QYZ) and Huitong (HT) experimental stations managed by the Chinese Academy of Sciences (CAS) and Central South University of Forestry and Technology (CSUFT), respectively. The QYZ station (115º04'E, 26º45'N) is located in Taihe county, Jiangxi province (Figure1, Table 1), surrounded by farmland, with no obviously anthropogenic mercury sources such as coal-fired power plants and metal smelters in 25 km around. The HT station (109º45'E, 26º50'N) is located in Huitong county, Hunan province, about 100 km away from the Wanshan Mercury Mine (WS), which used to be the largest mercury mine in China. The two study sites have the similar climate condition. The dominant soil and vegetation types (Table 1) are widely distributed in subtropical monsoon climate zone in south China. The subtropical evergreen coniferous forests have fairly thick canopy, even in winter.

### 2.2 Flux monitoring

The continuous monitoring system of GEM vertical concentration gradient over forest canopy included a Hg detector, two series of intake pipeline, and an automatically controlled valve system (Figure 2). The air sampling head and pipeline was arranged on the flux tower, while the valve system and mercury detector was set in the cabin near the flux tower. Two automatic GEM analyzers, model 2537X and 2537B (Tekran Instruments Inc.), with the same working principle and the detection limit (less than 0.1 ng m$^{-3}$, Gustin et al., 2013), were used at QYZ and HT site respectively. Air intakes were placed at two different heights (25 and 35 m of the 41 m-high flux tower at QYZ site; 22.5 and 30.5 m on the 33m-high flux tower at HT site). Considering the extremely large disturbance of temperature and wind speed over forest canopy, especially close to the canopy, the lower air intake should be set at least half canopy height (Table 1) above the canopy to ensure the stability of the results (Lindberg et al., 1998). Besides, all the air intakes would be fixed out of the tower body more than 1 m to avoid the influence of the tower. Passing a particulate filter membrane (0.2 μm) and a soda lime adsorption tank just after the intake to remove particulate matters, organic matters and acid gases, the in-gas from each height was pumped through a separated pipe (Φ = 0.25 in) to the same Hg detector in turn, controlled by two 3-way electromagnetic valves manipulated by a time relay. The electromagnetic valve switched once every 10 min, i.e., the measuring time of the gas from each height was 10 min, and it took 20 min for a whole measuring cycle. The design of the system including the pump ensured the continuing air flow at the same velocity in the two pipeline, whether the gas was sent to detect or no, to avoid the retention of air of the last cycle in the pipeline. The pipeline, air intakes and valves are made of Teflon to avoid the adsorption of Hg.

Meteorological parameters were also measured continuously by setting air temperature, humidity and wind speed sensors at the two heights (same to the air intakes), the solar radiation sensor and rainfall monitor at the higher height, and soil temperature and moisture sensors at 5 cm depth in soil about 20 m away from the flux tower. All the sensors adopted international advanced and reliable model (Table S1). All kinds of meteorological data were output by the data acquisition system (CR1000, Campbell Scientific Inc., USA) every five minutes.

The observations of GEM concentration gradient and meteorological parameters lasted for one year at both sites from January
to December in 2014.

**2.3 GEM flux calculation**

The dynamic Flux Chamber (DFCs) and micrometeorological techniques (MM) are the mostly widely applied approaches for
surface/atmosphere GEM flux quantification (Zhu et al., 2016). The MM methods, including of direct flux measurement
method (the relaxed eddy accumulation method, REA) and the gradient methods (further divided to the aerodynamic gradient
method, AGM, and the modified Bowen-ratio method, MBR), were usually defined to measure the GEM flux over forest
canopy with the advantages of no interference on measuring interface and high capability of chronical measuring large scale
fluxes. The AGM method, which has been used over grasslands, agricultural lands, salt marshes, landfills, and snow surface
(Lee et al., 2000; Kim et al., 2001; Kim et al., 2003; Cobbett et al., 2007; Cobbett and Van Heyst, 2007; Fritsche et al., 2008b;
Fritsche et al., 2008c; Baya and Van Heyst, 2010), was used in this study. According to the AGM method, the GEM fluxes
($F$, ng·m$^{-2}$·s$^{-1}$) over forest canopy was calculated on the basis of the measurement of the vertical concentration gradient by
using the following Eq. (1):

$$F = K\frac{\partial c}{\partial z}, \tag{1}$$

Where $K$ is turbulent transfer coefficient (m$^2$ s$^{-1}$), $c$ is GEM concentration in the atmosphere (ng m$^{-3}$), and $z$ is the vertical
height (m). Here, the GEM concentrations difference between the two air intakes divided by the height difference was assumed
to be the vertical gradient of atmospheric GEM concentration. Since the automatic GEM analyser switches between two gold
tubes and gets a value every 5 min, the two concentrations were averaged in each 10 min (matched to the single height sampling
interval by adjusting the time relay) to avoid possible bias caused by different gold tubes. The 20min variations of GEM
concentration at certain height were between -2% to 2% and -4% to 4% (95% confidence interval) at QYZ and HT sites
respectively. Thus, the GEM concentration was in a semi-steady state during the sampling interval. The GEM concentration
differences were calculated as the average concentrations at the higher height minus the two adjacent average concentrations
at the lower sampling height (all in 10 min interval). Thus, the vertical gradient of air GEM concentration can be gained every
10 min. Turbulent transfer coefficient $K$ was calculated through specific steps (Supplementary Information, SI) according to
the similarity theory after the measurement of the wind speed and temperature profile (Yu and Sun, 2006).

**2.4 Quality control**

In order to ensure the accuracy of the measurement results, regularly maintenance and calibration was performed to the
continuous monitoring system at both two sites. The particulate filter membrane on the air intake was changed once a week.
In addition, the soda-lime tank after the air intake and the filter membrane before the Hg analyzer was replaced monthly. The
automatic calibrations of the internal mercury source of Tekran 2537X and Tekran 2537B were done once every 24 h. The
manual calibration by placing the air intakes in certain Hg concentration (Tekran 2505, Tekran Inc.) for 24h were done once
every one month. The recovery rates were between 95 to 105% with an average value of 100.3%.
We did blank experiments, i.e., measuring the detection limit of the concentration gradient for the monitoring systems before
the installation, when the air intakes were both placed indoor with stable mercury concentration. It turned out that the
differences of GEM concentration between the pipelines were $0.004 \pm 0.017$ ng m$^{-3}$ and $0.010 \pm 0.024$ ng m$^{-3}$ ($n > 60$) at QYZ
and HT sites, respectively. The detection limit of the concentration gradient of the system was defined as the mean of detecting
difference results plus one standard deviation (Fritsche et al., 2008b). Therefore, the detection limits were 0.021 ng·m$^{-3}$ and
0.034 ng·m$^{-3}$ at QYZ and HT sites, respectively. It means that there was no significant difference between the two GEM
concentrations at different height when the discrepancy was lower than the detection limits in the field experiments. In addition,
the parallelity of the two pipelines in the system was detected every month by moving the air intakes to the cabin and run
continuously for at least 24 h. The pipeline need clean by soaking 24 h with 15% nitric acid then cleaning with ultrapure water
and acetone in turn, finally drying with zero mercury gas (Zero Air Tank, Tekran Inc.), until the difference of GEM
concentration between the two pipelines was less than 0.02 ng m$^{-3}$. There was a spare pipeline system at each site to avoid the
pause of monitoring due to pipeline cleaning. The blank experiments to measure the monitoring system error were conducted
before the installation by placing the air intakes in the zero mercury gas (Zero Air Tank, Tekran Inc.) for 48h. There were
almost no adsorption/emission from the monitoring system (including of the long Teflon tube, the soda-lime tank and the
electromagnetic valves) with the measurement results less than the detection limit of the instrument (0.1 ng m$^{-3}$).
The result measured by AGM represent a mean value of regional GEM flux, i.e, footprints area of tower, which is related to
the measuring height and meteorological conditions (Fritsche et al., 2008b). Previous study estimated that the footprint of
intake at 40 m height on the flux tower was 100 - 400 m (Zhao et al., 2005). Therefore, the footprints of the intakes located at
different height may be similar due to the relatively uniform distribution of *pinus massoniana* or *cunninghamia lanceolata*
forest within 500 m around the flux towers in our research.
The concentrations gradient lower than the system detection limit could not be truncated in case of the overestimation of GEM
flux when calculating the average GEM flux in previous studies (Fritsche et al., 2008b; Converse et al., 2010). The proportion
of the data which had the GEM concentration gradient larger than the detection limit in this study was larger than 85%, which
was higher than that in the previous study on grassland (about 50%; Fritsche et al., 2008b). The reason of such high quality
data might be the larger height difference (10m at QYZ site and 8m at HT site, vs. 2m in the grassland study), higher GEM
concentration, and larger exchange surface of forest than grassland. In accordance with the inaccurate measurement by AGM
under the high atmospheric stability (Converse et al., 2010), especially in temperature inversion, the calculation of turbulent
transfer coefficient $K$ could not converge, and the flux would be eliminated. In addition, the data would be eliminated when
the GEM flux exceed the range of the monthly mean $\pm$ 3 standard deviations, or during instrument failure and operation
instability.

## 3 Results and discussion

### 3.1 Hourly and daily variations in GEM concentrations and fluxes

QYZ and HT stations have both subtropical monsoon climate, with hot and rainy summers, and cold and dry winters (Table S2). Atmospheric GEM concentrations (the average concentration at two heights) were lower during spring and summer, and higher in winter and fall, with an annual average value of 3.64 ng m$^{-3}$ (1.89 ~ 6.26 ng m$^{-3}$, 5% ~ 95% confidence interval) at QYZ site (Figure 3), which was far higher than the mercury concentrations in background region in the Northern Hemisphere (1.5~2.0 ng·m$^{-3}$, Steffen et al., 2005; Kock et al., 2005; 1.51 ng·m$^{-3}$ in 2014, Sprovieri et al., 2016;) and correspond to the observed results in southeast China (2.7~5.4 ng·m$^{-3}$, Wang et al., 2014a). Although there were no major anthropogenic mercury emission sources near the QYZ station, the high concentration may be attributed to regional residential coal combustion (Wu et al., 2016) and high background GEM concentration in China (Fu et al., 2015). The annual average GEM concentration at HT station was 5.93 ng m$^{-3}$ (2.46 ~ 11.6 ng m$^{-3}$, 5% ~ 95% confidence interval), even higher than that at QYZ station.

The diurnal variation of fluxes indicated that the GEM flux increased gradually with the increase in air temperature and solar radiation in the daytime in all four seasons. The peak fluxes were averaged to 30.9, 29.3, 50.9 and 29.6 ng m$^{-2}$ h$^{-1}$ (22.6, 46.2, 46.2 and 44.7 ng m$^{-2}$ h$^{-1}$) in winter (December - February), spring (March - May), summer (June - August) and fall (September – November) respectively at QYZ (HT) at around 1:00 pm. In contrast, the GEM fluxes were stable at around zero or even negative at night, indicating a state of Hg balance at QYZ site and a strong sink at HT site. This pattern was similar to the Hg emission characteristics of soil (Ma et al., 2016), vegetation (Luo et al., 2016), and terrestrial surfaces (Stamenkovic et al., 2008). Modelling results of the diurnal variation of GEM fluxes over canopy for deciduous needle-leaf forest (Wang et al., 2016) also showed the similar trend.

A clear GEM absorption (negative fluxes) not only occurred at night but also in the morning in spring at both sites (Figure 4b). A small and a large depletion peaked at 9:00 am and 11:00 am at QYZ and HT sites, respectively in spring might result from the vegetation uptake, which was found by direct monitoring of GEM emission from foliage (Luo et al., 2016; Converse et al., 2010; Stamenkovic and Gustin, 2009). The daytime-GEM emission fluxes were significantly higher in summer and lower in winter with the changes of air temperature and solar radiation. With longer daytime and higher temperature, there were fewer hours per day in a state of GEM sink in summer (Figure 4c). The atmosphere-forest exchange of GEM became weaker in the fall as the decline in temperature and the dormant of plant growth (Figure 4d). There were also seasonal differences on diurnal variation of GEM emission from soil (Ma et al., 2016) and vegetation (Luo et al., 2016), with highest values occurring in summer, followed by spring and fall, while the lowest value in winter.

The two stations had the similar temperature due to the same climate condition and latitude (Table 1 and S2). Relatively higher value and later peak of solar radiation (except for summer) at HT site might result from the higher altitude and lower longitude, which would delay the peaks of emission flux in winter, spring, and fall. Relatively larger standard variance of GEM flux at HT site indicated the higher fluctuation, which might be ascribed to the fluctuating GEM concentration. HT station is close to WS Mercury Mine, the GEM concentration is vulnerable to the meteorological factors like wind direction.

## 3.2 Monthly variations in GEM concentrations and fluxes

The monthly mean value of GEM concentration seemed quite even throughout the year at both QYZ and HT Sites, with three peak values in January, June, and November (4.52, 4.32, and 4.25 ng m$^{-3}$ at QYZ site; 6.73, 6.74, and 7.14 ng m$^{-3}$ at HT site), and two bottom values of 2.33and 2.89 ng m$^{-3}$ (in March and July) at QYZ site and 4.29 and 3.34 ng m$^{-3}$ (in February and July) at HT site. In generally, monthly variations of fluxes exhibited an opposite trend of the concentration, almost all the larger fluxes emerged in the months with lower GEM concentration.

All the monthly mean GEM fluxes were positive at QYZ station (Figure 5), indicating that the forest was net atmospheric GEM source in each month. The relatively low GEM flux (3.13 ng m$^{-2}$ h$^{-1}$) and lowest air temperature (7.15 °C) occurred in December. The monthly mean GEM fluxes rapidly rose from December to March, coinciding with the increase in air temperature and solar radiation, followed by a sudden fall to 1.56 ng m$^{-2}$ h$^{-1}$ in April, and a slight increase to 4.40 ng m$^{-2}$ h$^{-1}$ in June. After that, the GEM flux rapidly increased to 11.5 ng m$^{-2}$ h$^{-1}$ in July and peaked at August (12.8 ng m$^{-2}$ h$^{-1}$), then gradually reduced to 6.84 ng m$^{-2}$ h$^{-1}$ in November, corresponding to the decrease in air temperature. Generally, the increase of solar radiation and air temperature would cause the increasing in GEM emission from soil and vegetation (see section 3.3). The monthly variations of annual Hg emission fluxes from forest soil in South Korea showed similar trend with air temperature (Han et al., 2016). Mainly affected by soil emissions, the changes of GEM fluxes showed similar trend as those of air temperature and solar radiation in winter and fall. In contrast, the GEM fluxes greatly decreased in the growing season, mainly influenced by vegetation uptake of GEM (Millhollen et al., 2006a; Stamenkovic and Gustin, 2009).

Different from QYZ station, the forest was a GEM sink in November, December and January with a negative value of monthly mean GEM flux of -6.82, -7.64, and -3.60 ng m$^{-2}$ h$^{-1}$ respectively at HT station (Figure 5). The monthly mean GEM fluxes gradually elevated and became positive in February to April, subsequently fell to negative in May. Then, coinciding with the change of air temperature, the GEM fluxes increased again, peaked in August (6.86 ng m$^{-2}$ h$^{-1}$), and gradually decreased to negative in November. Although monthly variation of GEM fluxes at HT site was similar to that at QYZ site, HT site had overall lower GEM fluxes but higher atmospheric GEM concentration than QYZ station. The annual average atmospheric mercury concentration at HT site was 62% higher than that at QYZ site (Table 1). Higher concentrations of atmospheric mercury would inhibit the Hg release from soil and plants, and increase the GEM absorption of foliage (see also in section 3.2). In addition to the influence of high atmospheric GEM concentration, the current-year foliage of *cunninghamia lanceolata* (dominant species at HT station, Table 1) have larger absorption than *pinus massoniana* at QYZ indicated by larger Hg content in needles and litters (Figure S3; Luo et al., 2016).

The monthly mean daytime-GEM fluxes always had positive values, which were much larger than the values at night (with small negative values in December, January, April and May, and near-zero in other months) at QYZ site (Figure 6). Thus, the GEM flux over forest canopy was mainly attributed to the emission during the daytime at QYZ site. The monthly mean GEM fluxes were also positive during the daytime but all negative at night at HT site. HT site had larger monthly mean emission

fluxes during the daytime and larger absorption fluxes at night (Figure 6). As a total effect, the monthly fluxes were lower than
those in QYZ (Figure 5).

**3.3 Factors influencing GEM flux**

In order to evaluate the influences of the environmental conditions and atmospheric GEM concentration on the GEM fluxes,
the correlation analysis between the flux and each factor had been calculated (Table 2). It showed that the GEM flux over
forest canopy was negatively correlated with atmospheric GEM concentration at both sites except in summer at QYZ station.
The inhibiting effect of atmospheric GEM concentration on GEM emission was not only reflected by the lower emission fluxes
at HT site comparing with those in QYZ site (Figure 5), but also by an instant decline in GEM flux after a sudden increase in
ambient GEM concentration. For instance, continuous measurement data during five typical days in each season (Figure 7)
showed an absorption peak on February 3 and May 5 at QYZ site and May 14 and August 24 at HT site caused by the increase
in air GEM concentrations. According to the wind direction records, the sudden rise of GEM concentration to 22.94 ng m$^{-3}$ on
May 14 and 21.21 ng m$^{-3}$August 24 at HT site might be caused by the approach of a high-mercury-content air mass from WS
Mercury Mine leading by northwest wind. Elevated ambient GEM concentration has been found to suppress GEM flux by
reducing the GEM concentration gradient at the interfacial surfaces (Xin and Gustin, 2007). At locations where ambient Hg
concentration is high, absorption (or deposition) is predominately observed despite of influence of meteorological factors
(Wang et al., 2007; Niu et al., 2011). Although the increase in GEM concentration would inhibit mercury emissions of foliage
and soil, the emission fluxes had positive correlation with atmospheric GEM concentration in summer (Figure S4) because the
large emission of GEM concentration in hot summer might result in an increase of air mercury concentration.
The GEM flux was positively correlated with solar radiation, air temperature, and wind speed at both QYZ and HT sites (Table
2). Solar radiation has been found to be highly positively correlated with soil and vegetation GEM flux (Carpi and Lindberg,
1997; Boudala et al., 2000; Zhang et al., 2001; Gustin et al., 2002; Poissant et al., 2004; Bahlmann et al., 2006), because it can
enhance Hg$^{2+}$ reduction and therefore facilitate GEM evasion (Gustin et al., 2002). For instance, there was a high GEM
emission peak at noon in winter (Figure 7; from February 1 to 3 at QYZ site and February 19 to 20 at HT site) even with
extremely low temperature. In addition to solar radiation, air temperature had significant effect on GEM flux, especially in
summer. Continued GEM emissions occurred in the daytime without strong solar radiation, or in the evening under the high
temperature in the summer (Figure 7; August 18 to 19 at QYZ site). Recent studies also showed that the GEM emission flux
from soil would be mainly controlled by the air temperature (Moore and Carpi, 2005; Bahlmann et al., 2006). Compared with
that in summer, GEM emission peak had decreased (Figure 7; 53.0 and 60.8 ng·m$^{-3}$ h$^{-1}$ on November 9 and 10 vs. 77.6 on
August 16 at QYZ site; 213, 206 and 103 ng·m$^{-3}$ h$^{-1}$ on November 15, 16 and 18 vs. 322 and 276 ng·m$^{-3}$ h$^{-1}$ on August 21 and
22 9 in HT site) on the sunny day in the fall due to the decrease in temperature. In addition, as wind speed increased, the air
turbulence on the surface of soil and foliage would speed up, and thus enhance the desorption of GEM on the interface
(Wallschlager et al., 2002; Gillis and Miller, 2000; Eckley et al., 2010; Lin et al., 2012), which may explain the positive
correlation between GEM flux and wind speed. Soil temperature mainly impacting on the emission of soil, and also showed

positive correlation with GEM fluxes except for in the winter with low soil temperature (Table 2). One possible explanation of the exception was that the change of soil temperature had no significant influence on the microbial activity and the reaction rate in soil if soil temperature was lower than a certain value (Corbett-Hains et al., 2012).

Air humidity generally was negatively correlated to the GEM flux over forest canopy (Table 2). Higher relative humidity may decrease stomatal conductance and thus lower transpiration of needles, which would result in decreases in GEM emissions (Luo et al., 2016). The correlation between GEM flux and soil moisture was not sure at QYZ station, e.g., positive in winter, negative in spring and fall, but no significance in summer. It seems that the influence of soil moisture on soil mercury emissions was uncertain, depends on the state soil water saturation (Figure S5). Previous studies supported that adding water to dry soil promotes Hg reduction, because water molecules likely replace soil GEM binding sites and facilitates GEM emission. However, Hg emission is suppressed in water saturated soil because the soil pore space filled with water hampers Hg mass transfer (Gillis and Miller, 2000; Gustin and Stamenkovic, 2005; Pannu et al., 2014). For instance, intensive soil GEM emission was synchronized to the rainfall at around 9:00 pm on August 16 and 8:00 pm on August 17 at QYZ site (Figure 7). In addition, the continuous but weaker rainfall from November 6 to 7 might also increase the GEM emission, in comparison with that in November 8 under the same solar radiation and temperature. Actually, continuous but weaker rainfall would lead to the increase of soil moisture, but not necessarily caused soil water saturation. Soil moisture content monitoring results had shown that the soil moisture content had a certain rise but remained below 0.28 during this period, which was lower than the highest value (0.52) during the annual monitoring. However, no significant emission flux was observed on August 19 after a series of strong rainfall. Repeated rewetting experiments showed a smaller increase in emission, implying GEM needs to be resupplied by means of reduction and dry deposition after a wetting event (Gustin and Stamenkovic, 2005; Song and Van Heyst, 2005; Eckley et al., 2011). The correlation between GEM flux and soil moisture was not significant in all of the seasons since the fluctuation of soil moisture content was small with the annual range of 0.21~0.34 at HT site, and the change of soil moisture content had far less impact on the soil GEM emissions.

The temporal variation of vegetation growth would play an important role in the forest GEM emission because of the vital function of vegetation to Hg cycle in forest ecosystem through changing environmental variables at ground surfaces (e.g., reducing solar radiation, temperature and friction velocity) (Gustin et al., 2004), and providing active surfaces for Hg uptake. Recent measurements suggested that air–surface exchange of GEM is largely bidirectional between air and plant, and that growing plants act as a net sink (Ericksen et al., 2003; Stamenkovic et al., 2008; Hartman et al., 2009). The negative exchange GEM fluxes at night at both two sites in this study should be mainly attributed to GEM adsorption by vegetation (Figure 6). In addition, GEM absorption capacity of foliage began to weaken at the end of growing season in November when the absorption peaks were smaller than that in spring at both QYZ and HT sites (Figure 7). The stomata open in the morning will also accelerate the forest absorption of Hg by vegetation, lead to the emergence of absorption peak even in the morning (Luo et al., 2016).

## 3.4 Forest as source/sink of GEM

GEM flux measurements over forest canopy indicated that QYZ forest at the mildly polluted site was a net source of GEM in all seasons, with the highest and lowest GEM emissions in summer (8.09 ng m$^{-2}$ h$^{-1}$) and spring (5.25 ng m$^{-2}$ h$^{-1}$, early growing season) respectively. In contact, significant differences in GEM fluxes were observed among seasons at HT, the moderately polluted site, indicating a clear sink in winter (dormant season), a slight source in spring and fall, and a large source in summer (Table 3). As the total effect, the forest ecosystem at HT site had a net GEM emission with a magnitude of 0.30 ng m$^{-2}$ h$^{-1}$ for a whole year. These results suggest that the subtropical forests in our study region should be the substantial GEM source, and the differences among seasons emphasized the importance of capturing GEM flux seasonality when determining total Hg budgets. As mentioned before, there was almost no difference of climate conditions between QYZ and HT sites, with the similar soil type and latitude, and little difference in the vegetation growth. However, the HT site with higher atmospheric GEM concentration had relatively lower GEM fluxes in all seasons in comparison with those in QYZ site. It emphasized again the importance of atmospheric GEM concentration on the GEM fluxes.

The GEM fluxes over forest canopy were the sum of emission fluxes from soil and vegetation, and extremely difficult to quantify. GEM exchange of foliage/atmosphere or soil/atmosphere is both bi-directional, with net adsorption occurring at elevated air Hg concentration while net emission when typical ambient concentration was lower than the "compensation point" (Converse et al., 2010; Ericksen et al., 2003; Stamenkovic et al., 2008; Hartman et al., 2009). However, the study of foliage/atmosphere mercury exchange at QYZ indicated that the vegetation presented as a net GEM source as the total effects with a value of 1.32 ng m$^{-2}$ h$^{-1}$ (2.19, 0.32, 2.51 and -0.01 ng m$^{-2}$ h$^{-1}$ in winter, spring, summer and fall respectively) caused by high rates of photoreduction and plant transpiration due to high temperature and radiation, relatively large leaf surface area and elevated mercury deposition, but a clear sink in the growing season with stomatal opening (Luo et al., 2016) even under the relatively lower atmospheric GEM concentration. In addition, the study of the mercury exchange between atmosphere and soil under the forest canopy at QYZ using the DFC methods also showed the soil manifested as net GEM sources at all the seasons (Figure S6, 0.13 ± 0.43, 1.54 ± 1.78, 4.76 ± 1.86 and 2.07 ± 1.73 ng m$^{-2}$ h$^{-1}$ in winter, spring, summer and fall, respectively; unpublished data). Thus, the net emissions observed at QYZ were contributed by both soil and foliar emissions. The GEM fluxes over forest canopy (8.09 ng m$^{-2}$ h$^{-1}$) in this study were almost similar to the sum (7.27 ng m-2 h-1) of emission fluxes from foliage and soil in summer, but had lager values in other seasons. It might be because of the underestimation of the GEM fluxes from soil due to the decreased turbulence in chamber using the DFC method, and the lack of GEM fluxes from the undergrowth vegetation. Although the foliage/atmosphere and soil/atmosphere mercury exchange at HT have not been measured, respectively, the comparison of Hg content of current-year foliage and soil between two sites might indicate that there were larger GEM emission fluxes from soil but much larger GEM adsorption by foliage. Until now, there are merely few researches using AGM to monitor the GEM flux above forest canopy even in short period. Previous studies showed that the exchange fluxes of GEM vary in sign and magnitude (Table 3). Lindberg et al. (1998) measured GEM fluxes over a mature deciduous forest, a yang pine plantation, and a boreal forest floor using the MBR method and suggested that global forest is a

net source of GEM with an emission of 10-330, 17-86 and 1-4 ng m$^{-2}$ h$^{-1}$ at daytime, respectively (Table 3). The observation
of Hg fluxes in a deciduous forest using a REA method showed a net GEM emission of 21.9 ng m$^{-2}$ h$^{-1}$ during summer (Bash
and Miller, 2008). However, a study in Québec, Canada showed that GEM concentrations at a maple forest site are consistently
lower than those measured at an adjacent open site, indicating a Hg sink for the forest (Poissant et al., 2008). Similarly, the
lower GEM concentrations observed in leaf-growing season at forest sites across the Atmospheric Mercury Network (AMNet)
in USA (Lan et al., 2012), Coventry Connecticut, England (Bash and Miller, 2009), Mt. Changbai, Northeast China (Fu et al.,
2016) also suggest forest as a net GEM sink during the growing season. Different results were obtained by AGM and MBR
method at the same time (Converse et al., 2010) (Table 3). There was limiting comparability of fluxes data reported in literature
because of the lack of a standard method protocol for GEM flux quantification (Gustin, 2011; Zhu et al., 2015). The
discrepancy in the measured GEM exchanges between forest and atmosphere is partially attributed to the uncertainties of the
flux quantification method (Sommar et al., 2013), but the forest structure, climate condition, background Hg concentration,
and forest soil Hg content could play critical roles in GEM emission from forest ecosystem. Unlike deciduous forest as a sink
of GEM in most previous studies, the evergreen foliage with relatively higher LAI at all seasons in the subtropical forests in
this study (in spite of the seasonal variations of vegetation growth) was demonstrated as a net GEM source to the atmosphere
(Luo et al., 2016). Evergreen tree species generally have higher exchange capabilities of GEM relative to deciduous tree species
and result in high rates of photoreduction and plant transpiration under the high temperature, solar radiation and soil Hg content.
In addition, extremely high soil Hg content (42.6 and 167 ng g$^{-1}$ at QYZ and HT sites shown in Table 1, while 63 ng g$^{-1}$ in in
Québec, Canada; Poissant et al., 2008) result from long-term elevated Hg deposition, the high temperature and solar radiation
would also contribute the net emission flux of GEM from forest soil in subtropical, south China. However, the observations in
this study were not higher than the results in the forests as GEM sources in previous studies, possibly due to the higher ambient
GEM concentration (3.64 and 5.93 ng m$^{-3}$ at QYZ and HT sites vs. 2.23 ng m$^{-3}$ in Tennessee, USA and 1.34 in Connecticut,
USA; Table 3). Although there were net GEM emissions (58.5 μg m$^{-2}$ yr$^{-1}$) from forest in this study at QYZ site based on the
measurement of the GEM fluxes over forest canopy, on account of extremely large Hg deposition (wet deposition:14.4 μg m$^{-}$
$^{2}$ yr$^{-1}$; dry deposition: 52.5 μg m$^{-2}$ yr$^{-1}$; Luo et al., 2016), the forest presented as a Hg source, overall.
**4 Conclusions and implication**
The high quality direct observation data of a mildly polluted and a moderately polluted site with typical climate, vegetation
type and soil type in south China could be important for implications for the regional Hg cycling estimation, and the awareness
of the role of forests in the global mercury cycle. From continuously quantitative MM-flux measurements covering wide
temporal scales at QYZ and HT sites in subtropical south China, it is inferred that forest ecosystems can represent a net GEM
source with the average magnitudes of 6.67 and 1.21 ng m$^{-2}$ h$^{-1}$ for a full year at a mildly polluted site (QYZ) and a moderately
polluted site (HT), respectively. GEM flux measurements were net source in all seasons at the mildly polluted site, with the
highest in summer because of the relatively high air temperature and radiation, and lowest in spring result from the vegetation

growth. For the moderately polluted site, a net sink occurred in the winter, a significant source in summer, and no significant flux during spring and fall. The GEM emission dominated in the daytime, and peaked at around 1:00 pm, while the forest served as a GEM sink or balance at night. It is worth noting that there was a lower emission fluxes of GEM at the moderately polluted site result from similar or even higher emission fluxes during daytime, but much higher adsorption fluxes at night than the mildly polluted site under the similar meteorological conditions. Although, the larger Hg content in soil would enhance the emission of soil and vegetation, the elevated GEM concentration suppresses the Hg emission, and increase the absorption by vegetation at the moderately polluted site. The result indicated that the atmospheric GEM concentration play an importance role in inhibiting the GEM fluxes between forest and air, coinciding with the negative correlation between GEM fluxes and atmospheric GEM concentration. In addition, the forest should be pay attention as a critical increasing source with the decline atmospheric GEM concentration because the Hg emission abatement in the future, and the increasing emission might result from the re-emission of legacy Hg stored in the forest.

The GEM flux over forest canopy was the sum emission flux of soil and vegetation, and showed monthly variations caused by the temporal variation of vegetation growth, atmospheric GEM concentration and meteorological conditions including of air temperature, radiation and wind speed. The correlation between GEM fluxes and factors had been analysed, combined with the characteristics of GEM exchange between soil (or foliage) and air. It indicated that GEM fluxes were positively correlated with air temperature, soil temperature, wind speed, and solar radiation, but negatively correlated with air humidity. The influence of soil moisture content was uncertain, depends on whether the soil water saturated and the initial state of the soil. In addition, vegetation growth would play an important role in the decline in forest GEM emission in spring. The difference in climate conditions and ambient GEM concentration should be considered when estimating the global forest GEM emission.

**Acknowledgement**

The authors are grateful for the financial support of the National Basic Research Program of China (No. 2013CB430000) and the National Natural Science Foundation of China (No. 21377064 and No. 21221004). The authors also greatly acknowledge the supports from Qianyanzhou Forest Experimental Station and Huitong Forest Experimental Station, and the help in system maintenance from Yuanfen Huang and Yungui Yang.

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

**Table1.** Description of QYZ and HT experimental station

| Station sites | QYZ | HT |
|---|---|---|
| Location | 115º04'E, 26º45'N | 109º45'E, 26º50'N |
| Administrative region | Guanxi town, Jiangxi province | Guangping town, Hunan province |
| Altitude (m) | 30~60 | 280~390 |
| Climate type | Humid subtropical monsoon climate | |
| Mean annual temperature (°C)[a] | 18.6 | 15.8 |
| Mean annual precipitation (mm)[a] | 1361 | 1200 |
| Dominated tree species (relative abundance) | *Pinus massoniana* (86.5%) | *Cunninghamia lanceolata* (92.4%) |
| Other predominant vegetative species | *Pinus elliottii; Quercus fabei; Vitex negundo; Rhododendron plonch; Ischaemum indicum* | *Marsa japonica ; Ilex pirpurea; Cyclosorus parasticus; Woodwardia prolifera* |
| Forest age | 31 | 27 |
| Canopy height (m) | 16 | 14 |
| Leaf area index (LAI) in summer | 4.31 | 7.00 |
| Canopy density | 0.7 | 0.8 |
| Radiation transfer under canopy | 3.0% | 2.7% |
| Dominant soil type (Chinese soil name) | Udic Ferrisols (Red Earth) | Haplic Acrisol (Yellow Earth) |
| Organic matter content in surface soil (g kg$^{-1}$)[a] | 10~15 | 28.3 |
| Soil pH[a] | 4.52 | 3.85 |

| | | |
|---|---|---|
| Annual average GEM concentration (ng m$^{-3}$) [b] | 3.64 ± 1.82 | 5.93 ± 3.16 |
| Hg content in soil organic layer (ng g$^{-1}$)[c] | 76.2 ± 6.0 | 153 ± 28 |
| Hg content in surface (0~5 cm) soil (ng g$^{-1}$)[c] | 42.6 ± 2.3 | 167 ± 32 |

[a] Data of QYZ and HT stations according to Gao et al. (2014) and Wang et al. (2009), respectively;
[b] Mean value of the measurements at the height of 25 m and 35 m at QYZ site, 22.5 and 30.5 m at HT site;
[c] Analyzed based on 18 samples using a direct Hg analyzer (DMA80, Milestone Inc., Italy).


**Table 2.** Pearson's correlation coefficient between GEM flux over forest canopy and atmospheric GEM concentration or each environmental factor.

| Factors | Sites | Winter | Spring | Summer | Fall |
|---|---|---|---|---|---|
| GEM concentration | QYZ | -0.142** | -0.155** | 0.014 | -0.141** |
| | HT | -0.232** | -0.226** | -0.197** | -0.183** |
| Air temperature | QYZ | 0.272** | 0.166** | 0.31** | 0.298** |
| | HT | 0.143** | 0.121** | 0.188** | 0.135** |
| Air humidity | QYZ | -0.314** | -0.003 | -0.293** | -0.339** |
| | HT | -0.101* | -0.149** | -0.246** | -0.255** |
| Wind speed | QYZ | 0.159** | 0.176** | 0.162** | 0.166** |
| | HT | 0.119** | 0.180** | 0.106** | 0.162** |
| Soil temperature | QYZ | 0.025 | 0.165** | 0.288** | 0.175** |
| | HT | 0.015 | 0.174** | 0.253** | 0.201** |
| soil moisture | QYZ | 0.102** | -0.198** | 0.03 | -0.106** |
| | HT | 0.001 | -0.032 | -0.003 | 0.034 |
| Radiation | QYZ | 0.628** | 0.403** | 0.401** | 0.209** |
| | HT | 0.265** | 0.212** | 0.313** | 0.201** |

* Significant at $p < 0.01$ level;
** Significant at $p < 0.001$ level.

**Table 3.** Comparison of the GEM flux (ng·m$^{-2}$·h$^{-1}$) from different the observations.

| Vegetation type | Location | winter | spring | summer | fall | GEM con | method | Data source |
|---|---|---|---|---|---|---|---|---|
| Subtropical coniferous forest | Jiangxi province, China | 5.49 | 5.25 | 8.09 | 7.86 | 3.64 | AGM | QYZ site |
| | Hunan province, China | -3.62 | 0.83 | 4.40 | -0.40 | 5.93 | AGM | HT site |
| Mature hardwood | | – | – | 10-330 | – | 2.23 | MBR | |
| Yang pine plantation | Tennessee, USA | – | – | – | 17-86 | 1.45 | MBR | Lindberg et al. (1998)[a] |
| Boreal forest | Lake Gardsjon, Sweden | – | – | 1-4 | – | 2.02 | MBR | |
| Deciduous forest | Connecticut, USA | – | – | 21.9 | – | 1.34 | REA | Bash and Miller (2008) [b] |
| | Coventry Connecticut, England | – | – | -1.54 | – | 1.41 | REA | Bash and Miller (2009) |
| Meadow | Fruebuel,central Switzerland | 4.1 | -4.8 | 2.5 | 0.3 | 1.29 | AGM | Converse et al. (2010) |
| | | -2.9 | -1.5 | 3.2 | -3.0 | 1.29 | MBR | |

[a] mean value (90% confidence interval), only measured during daytime;
[b] median value of TGM (total gaseous mercury) flux


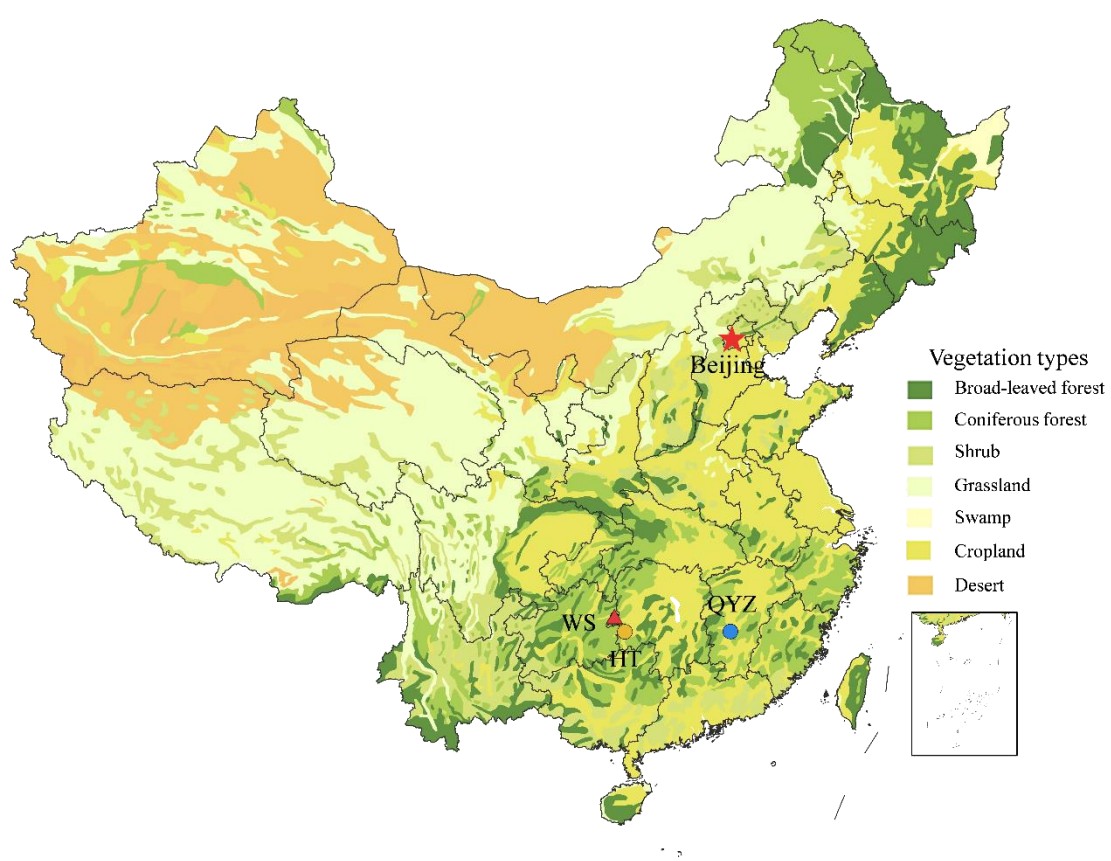

**Figure 1: Locations of the QYZ station, HT station and WS Mercury Mine. Vegetation map of China (CAS., 2007) as background.**

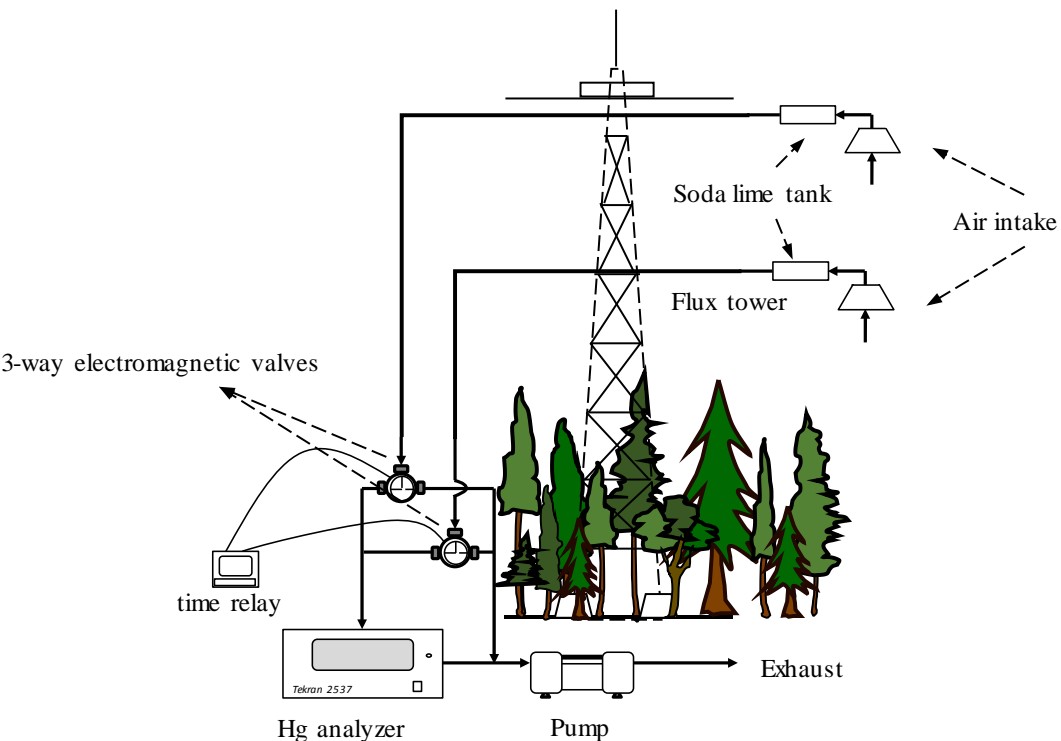

Soda lime tank

Air intake

Flux tower

3-way electromagnetic valves

time relay

Tekran 2537

Hg analyzer

Pump

Exhaust


**Figure 2: Apparatus used to monitor vertical concentration gradient of GEM above forest canopy**



(a)

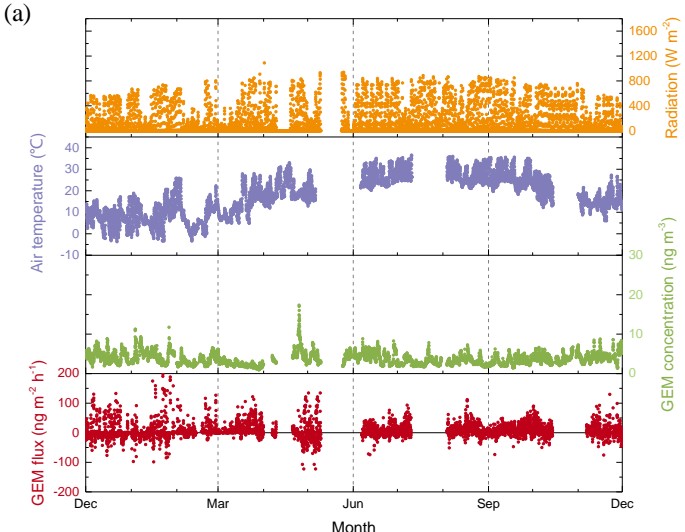


(b)

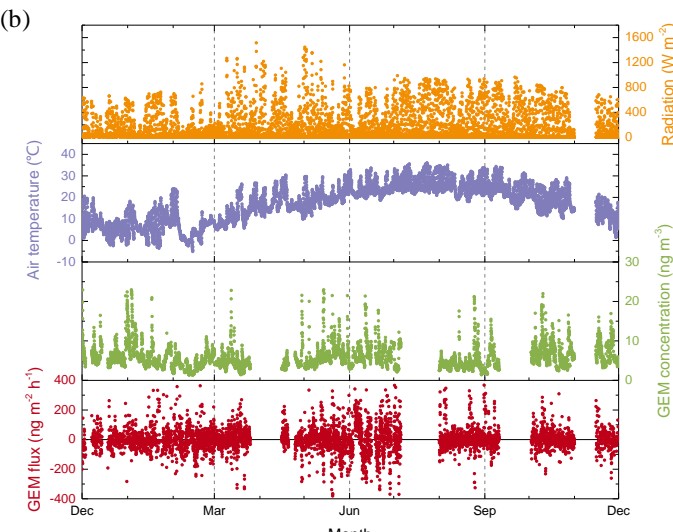



**Figure 3: Annual variations of solar radiation, air temperature, GEM concentration (the average value of the GEM concentration at two heights), and GEM fluxes at QYZ (a) and HT (b) stations. The observations lasted for one year at both sites (January to December in 2014). The data in April, May and December was supplemented with the data in 2013 due to the use of mercury analyzer for measuring the soil and vegetation emission at HT site. Data loss were caused by elimination of the values outside the range of the monthly mean ± 3 standard deviations, and the problematic data during the high atmospheric stability, instrument failure and instability operation. The annual variations of GEM gradient and turbulent transfer coefficient (K) was showed in Figure S1.**


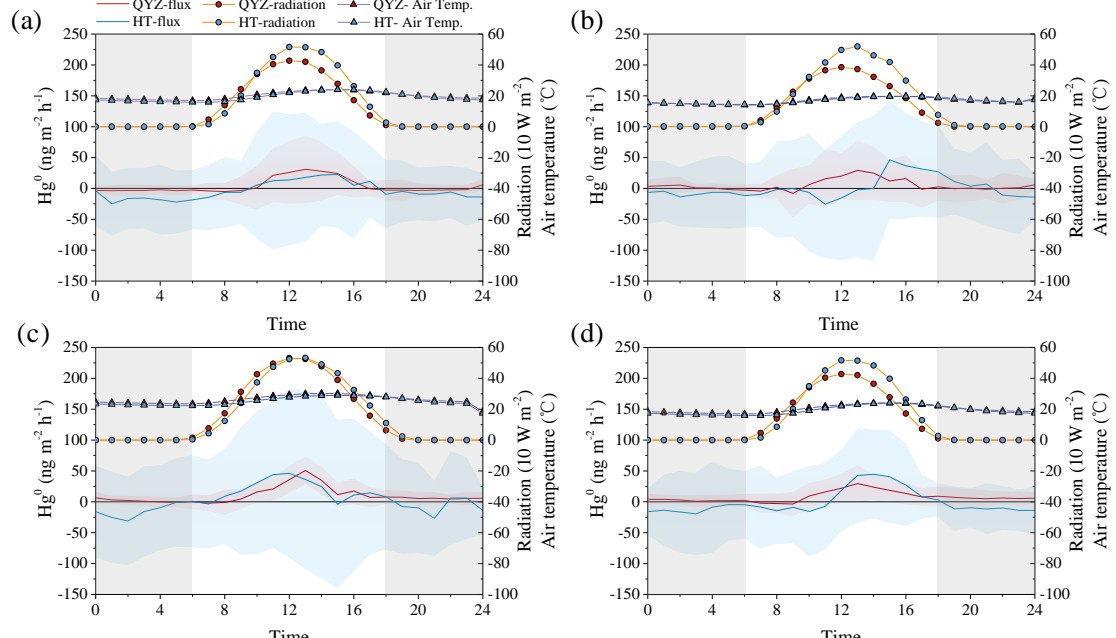


**Figure 4: Diurnal variation in GEM fluxes, air temperature and solar radiation over forest canopy in each season. (a) Winter:**
**December to February; (b) Spring: March to May; (c) Summer: June to August; (d) Fall: September to October. Lines and**
**envelopes depict mean values and standard variances. Diurnal variation in GEM gradient and turbulent transfer coefficient (K) in**
**each season at two sites was presented in Figure S2.**

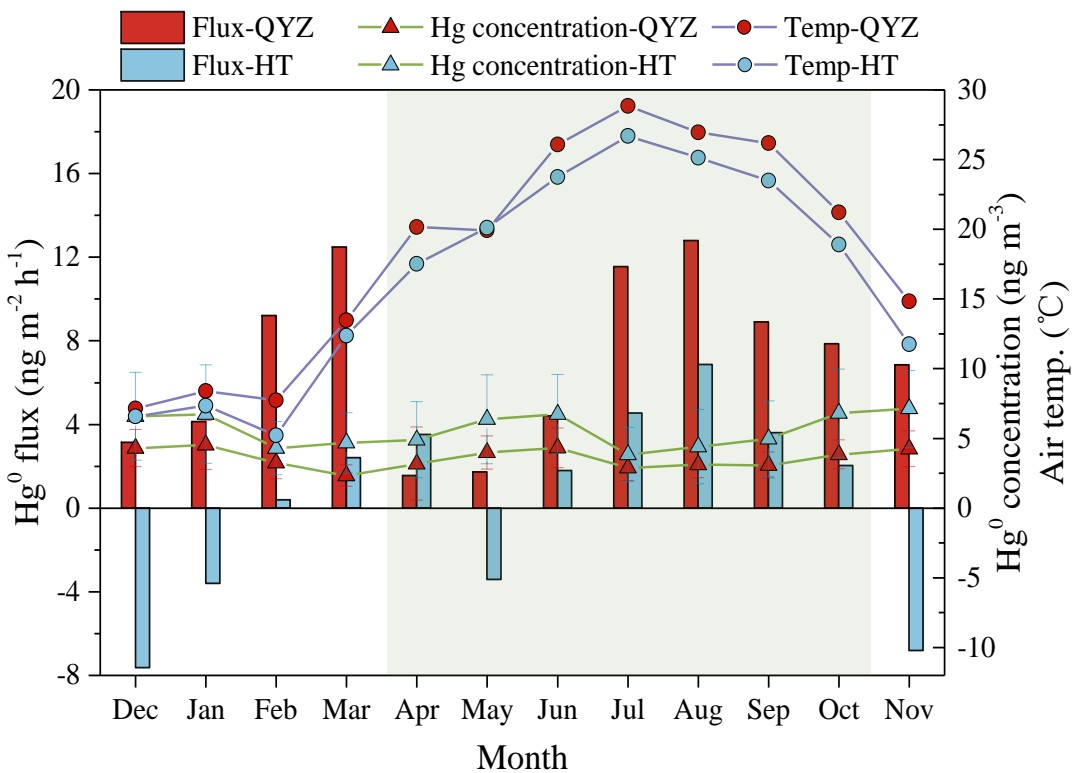


**Figure 5: Monthly variations of GEM flux, GEM concentration and air temperature at QYZ and HT sites. Leaf-growing season**

**was marked as the shaded area.**

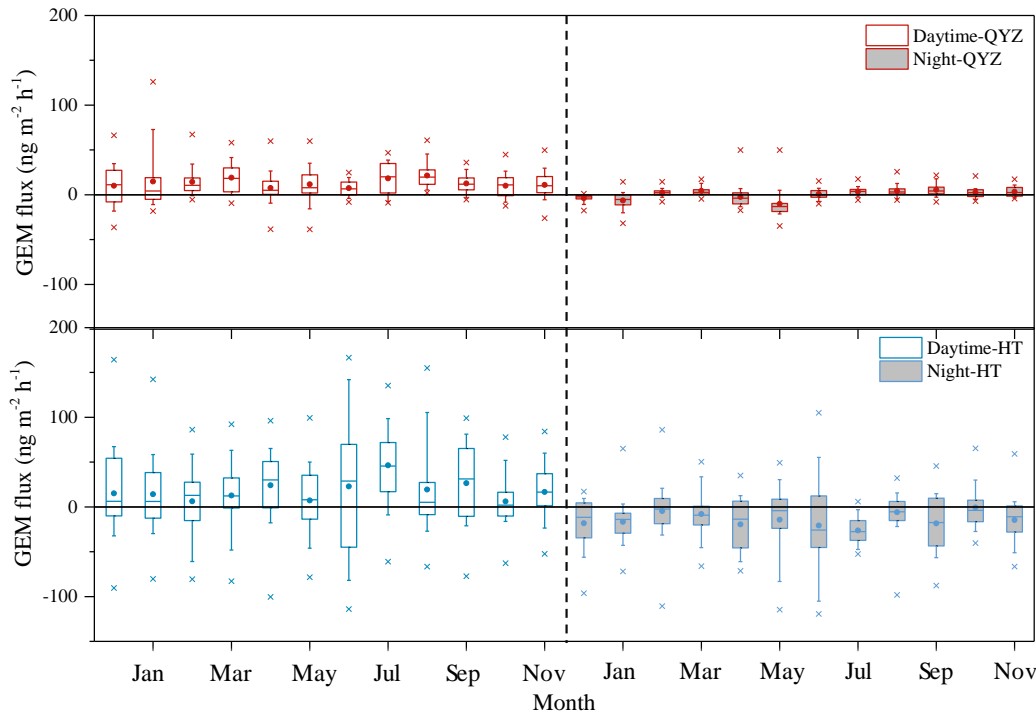


**Figure 6: Monthly variation in daytime GEM flux (upper panels) and night GEM flux (under panels) during the measurement**

**periods at QYZ (a) and HT (b) sites. Box horizontal border lines represent the 25th, 50th and 75th percentiles from bottom to top,**

**the whiskers include the 10th and 90th percentiles, and the outliers (cross) encompass the minimum and maximum percentiles. The**

**solid circle in the box represents the mean value.**


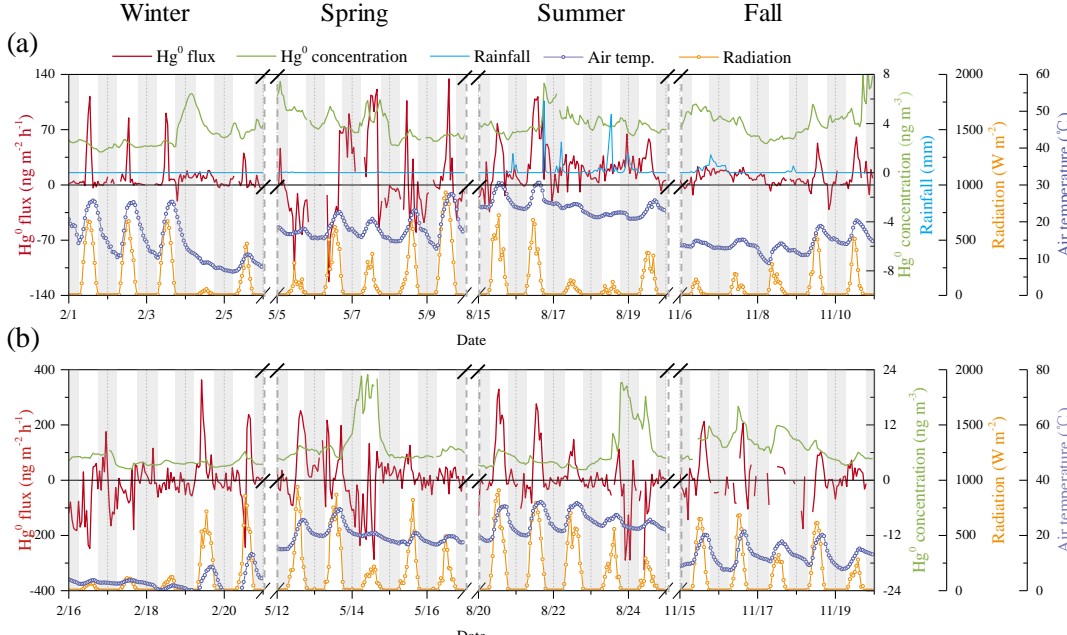


**Figure 7: The GEM flux, concentration and environmental conditions in some typical days in each season at QYZ (a) and HT (b)**


**sites. Dates refer to China Standard Time (major ticks indicate midnight). All the data were indicated one-hour average.**

