# Peer review of "Gaseous elemental mercury (GEM) fluxes over canopy of two typical"

_Atmospheric Chemistry and Physics, 2017_

## Referee Comment (RC1) · Anonymous Referee #1 · 10 Aug 2017

The manuscript reports flux data measured by the aerodynamic gradient (AGM) method over the canopy at two low-land forest sites in South China over a one-year period. Although the context of this study does not introduce new science beyond what has been known, such long-term flux datasets in forest ecosystem are rare and deserve consideration for publication. The manuscript is organized, read well and carefully drafted given the data that it presents. The data reporting sections (Section 3.1 – Section 3.4) are appropriately supported by data and the depth of conclusions/implication can benefit from incorporating the recent findings obtained from mass balance study and stable mercury isotope investigation in forest ecosystems. Overall, I recommend publication of the manuscript and have the following minor comments. 1. The LAI

and other descriptive characterization (e.g., other predominant vegetative species, forest floor coverage, radiation transfer under canopy) of the two forest sites should be provided for better assessment of the forest site. 2. Appropriately determining the turbulent transfer coefficient (K) is critical for estimating AGM flux, yet it is not clear in the manuscript how K varies. It will be useful to report the estimated K values and its diurnal variation in different in different season for evaluation of the reasonableness of estimated K values. 3. The quality control statement of the AGM flux measurement is limited only to the detection of Hg vapor but did not consider other sources of bias (the long tubing, flow synchronization, intermittent sampling at the two level, etc.) that might introduce uncertainty to the flux measurement. Would it be a possibility that these variables can also be assessed to better represent the measurement uncertainty? 4. The characterization of the two sites (clean and contaminated) could cause confusion. Given the mean annual TGM concentrations (3.64 and 5.93 ng m-3), both locations should have been considered under the influence of regional Hg emission plumes. I suggest using "mildly polluted" and "moderately polluted" to avoid the confusion. 5. It would have been extremely useful if Hg flux was also measured over the forest floor under the canopy. Such data will help enhance the source/sink discussion of Section 3.4. 6. Recent characterization of stable mercury isotopes in foliage of various ages and litter, along with the quantification of litterfall production and Hg deposition through litterfall all indicate that forest ecosystem is a net sink for Hg in remote regions. Such data are also supported by the vertical gradient measurement of TGM concentration from forest floor through well above forest canopy, showing increasing TGM concentration with sampling height. The primary conclusions of this study appear to contradict these recent findings, even given the elevated TGM concentration in air. Since the flux data above forest floor are not reported in the manuscript, it is not possible to determine if it is the re-emission of contaminated soil that leads to the overall Hg source term. An in-depth discussion regarding these discrepancy will significantly enhance the scientific value of the manuscript.
* * *

---

## Short Comment (SC1) · 1 Sep 2017

Forest is an important ecosystem on the earth. Characterizing the role of forest in the global biogeochemistry cycling of Hg is an important research topic in global Hg cycling studies. This study investigated the GEM gradient at two typical forests in subtropical zone of China. GEM concentrations and GEM fluxes measured are valuable for the atmospheric Hg studies in regional and global scale. I don't sea major errors in the sampling techniques and method, and the discussions, for most cases, are sound throughout the whole manuscript. I would suggest a publication after addressing the following questions.

[Figure]

Major points: The result from this study found the forest in subtropical zone of China is a net source of atmospheric GEM, and I have no doubt for this result. My question is, why the net emissions were observed in the study areas, soil emissions or foliar emissions? The authors did discuss this scientific issue in section 3.4, but we still don't know the exact causes. I think this is critical for atmospheric science. As we know, previous field observations on foliage/atmospheric Hg fluxes mostly revealed a net sink of GEM. If forest canopy is a net source in the study area, this would be an important finding. If the net emissions of GEM from forest were caused by elevated soil Hg emissions (soil Hg concentrations were elevated), then future studies regarding the mass balance of GEM in forest using litterfall approach approaches should also consider the soil emissions or reemissions. The net emissions observed in this study might be also due to many other factors including the specific tree species (evergreen tree species generally have higher uptake capabilities of GEM relative to deciduous tree species), lower leaf area index, reemission of dew water and transpiration stream, which should be also assessed in the manuscript.

Minor points: Line 79: why did the authors select the two sampling heights at QYZ and HT? Are you any pervious studies to support your setting; Will the chose of different sampling height affect the flux result? Line 81: 'half canopy height'? The lower sampling heights were 25 and 22.5 m agl, which were much higher than the canopy height. Section 3.1: I think the annual variations of GEM gradient at QYZ and HT are also important and should be presented in Figures. The authors should also show the annual, seasonal and diurnal trend of GEM gradient. Line 155: the authors should note the sampling height of the annual mean GEM concentration. Line 156: the global and northern hemisphere background should be referred to GMOS studies. Line 159: References are needed here Line 161: Do you have any evidence for this hypothesis? I strongly suggest the authors analyze the source-receptor relationships at the sampling sites? Measurement of GEM in the atmosphere is also an negligible part of this study. Line 190: should be 'have positive values at QYZ'. Line 196: References are needed here. Line 227: the source from WS mercury mining area? Any evidence? Line 297:

the study only reveal the whole forest is a net source, but not vegetation, you did not measure foliage/atmospheric Hg flux. Table-1: the relative abundance of major tree species should be listed.

---

## Author Comment (AC2) · 20 Sep 2017

Responses to Dr. Xinbin Feng

Thanks to Dr. Feng for giving us very useful comments to improve the manuscript. Detailed responses to the comments are list bellow (underlined).

Forest is an important ecosystem on the earth. Characterizing the role of forest in the global biogeochemistry cycling of Hg is an important research topic in global Hg cycling studies. This study investigated the GEM gradient at two typical forests in subtropical zone of China. GEM concentrations and GEM fluxes measured are valuable for the

atmospheric Hg studies in regional and global scale. I don't see major errors in the sampling techniques and method, and the discussions, for most cases, are sound throughout the whole manuscript. I would suggest a publication after addressing the following questions.

Major points: The result from this study found the forest in subtropical zone of China is a net source of atmospheric GEM, and I have no doubt for this result. My question is, why the net emissions were observed in the study areas, soil emissions or foliar emissions? The authors did discuss this scientific issue in section 3.4, but we still don't know the exact causes. I think this is critical for atmospheric science. As we know, previous field observations on foliage/atmospheric Hg fluxes mostly revealed a net sink of GEM. If forest canopy is a net source in the study area, this would be an important finding. If the net emissions of GEM from forest were caused by elevated soil Hg emissions (soil Hg concentrations were elevated), then future studies regarding the mass balance of GEM in forest using litterfall approach approaches should also consider the soil emissions or reemissions. The net emissions observed in this study might be also due to many other factors including the specific tree species (evergreen tree species generally have higher uptake capabilities of GEM relative to deciduous tree species), lower leaf area index, reemission of dew water and transpiration stream, which should be also assessed in the manuscript.

There are ongoing debates regarding whether or not forest is a sink or a source of GEM because the forest/air exchange flux is the sum of vegetation and soil exchange flux, depending on not only atmospheric concentration and meteorological conditions, but also plant community composition and soil structure (Bash and Miller, 2009; Converse et al., 2010) over shorter or longer periods. The study of foliage/atmosphere mercury exchange at QYZ indicated that the vegetation presented as a net GEM source, with a value of 1.32 ng m-2 h-1 (2.19, 0.32 2.51, -0.01, 1.32 ng m-2 h-1 in winter, spring, summer, and fall, respectively), caused by high rates of photoreduction and plant transpiration due to high temperature and radiation, relatively large leaf surface area, and

elevated mercury deposition (Luo et al., 2016). In addition, the study of the mercury exchange between atmosphere and soil under the forest canopy at QYZ through the DFC methods showed the soil also manifest as net GEM sources at all the seasons ($0.13 \pm 0.43$, $1.54 \pm 1.78$, $4.76 \pm 1.86$ and $2.07 \pm 1.73$ ng m-2 h-1 in winter, spring, summer, and fall respectivley; unpublished data). The above results were added in SI. Thus, the net emissions observed at QYZ were contributed by both soil and foliar emissions. The GEM fluxes over forest canopy (8.09 ng m-2 h-1) in this study were almost similar to the sum (7.27 ng m-2 h-1) of emission fluxes from foliage and soil in summer, but had lager values in other seasons. It might be because of the under-estimation of the GEM fluxes from soil due to the decreased turbulence in chamber using the DFC method, and the lack of GEM fluxes from the undergrowth vegetation. Although there were net GEM emissions (58.5 $\mu$g m-2 yr-1) from forest in this study at QYZ site based on the measurement of the GEM fluxes over forest canopy, on account of extremely large Hg deposition (wet deposition:14.4 $\mu$g m-2 yr-1; dry deposition: 52.5 $\mu$g m-2 yr-1; Luo et al., 2016), the forest presented as a Hg source, overall. It is a pity that foliage/atmosphere and soil/atmosphere mercury exchange at HT have not been measured, respectively. However, the comparison of Hg content of current-year foliage and soil between two sites might indicate that there were larger GEM emission fluxes from soil but much larger GEM adsorption by foliage. In addition to the lager GEM con-centration and soil Hg content, the in-depth discussion of discrepancy caused by many other factors including the flux quantification method, specific tree species (evergreen tree species generally have higher exchange capabilities of GEM relative to deciduous tree species), reemission from photoreduction and transpiration was added in Section 3.4.

Minor points: Line 79: why did the authors select the two sampling heights at QYZ and HT? Are you any pervious studies to support your setting; Will the chose of dif-ferent sampling height affect the flux result? Line 81: 'half canopy height'? The lower sampling heights were 25 and 22.5 m agl, which were much higher than the canopy height.

The result measured by AGM represent a mean value of GEM flux in regional area, i.e, footprints area of tower, which is related to the measuring height and meteorological conditions (Fritsche et al., 2008b). Previous study estimated that the footprint of intake at 40 m height on the flux tower was 100 - 400 m (Zhao et al., 2005). Therefore, the footprints of the intakes located at different height in our study may be similar due to the relatively uniform distribution of pinus massoniana or cunninghamia lanceolata forest within 500 m around the flux towers in our research.

Considering the extremely large disturbance of temperature and wind speed over forest canopy, especially close to the canopy, the lower air intake should be set at least half canopy height (Table 1) above the canopy to ensure the stability of the results (Lindberg et al., 1998). Thus, the lower sampling heights were about 1.5 times of the canopy height.

Section 3.1: I think the annual variations of GEM gradient at QYZ and HT are also important and should be presented in Figures. The authors should also show the annual, seasonal and diurnal trend of GEM gradient.

The GEM gradient and the turbulent transfer coefficient (K) are both critical for estimating GEM flux, and the annual, seasonal and diurnal variation in different season were presented in SI.

Line 155: the authors should note the sampling height of the annual mean GEM concentration.

The atmospheric GEM concentrations presented in manuscript were the average GEM concentrations at two heights, which was clarified in the revised manuscript.

Line 156: the global and northern hemisphere background should be referred to GMOS studies.

A reference for GMOS studies was added.

Line 159: References are needed here

Two references were added.

Line 161: Do you have any evidence for this hypothesis? I strongly suggest the authors analyze the source-receptor relationships at the sampling sites? Measurement of GEM in the atmosphere is also an negligible part of this study.

The sentence was removed. The source-receptor relationships at QYZ site was analyzed by using the Hg content of precipitation and throughfall, and the fluxes of soil/atmosphere, foliage/atmosphere and forest/atmosphere, the discussion was added in Section 3.4.

Line 190: should be 'have positive values at QYZ'.

Revised

Line 196: References are needed here.

"See section 3.3" was added because there were many related references and discussion in section 3.3.

Line 227: the source from WS mercury mining area? Any evidence?

The WS mercury mining area is located in the northwest of the HT site about 100 km away. The sudden rise of GEM concentration not only on May 14 presented in Figure 7, but also on January 17, September 5, October 10 and November 17 24 at HT site, corresponded to northwest wind prevailed according to the wind direction records. Thus, we believe that the sudden rise of GEM concentration might be caused by the approach of a high-mercury-content air mass from WS Mercury Mine leading by northwest wind.

Line 297: the study only reveal the whole forest is a net source, but not vegetation, you did not measure foliage/atmospheric Hg flux. Table-1: the relative abundance of major tree species should be listed.

The study of foliage/atmosphere Hg fluxes was also conducted at QYZ site by using

the DFC method. And the results were published in 2016, and could be used to explain the contribution of foliage to the whole forest GEM emission. The relative abundance of major tree species was listed in Table 1.

Please also note the supplement to this comment:
https://www.atmos-chem-phys-discuss.net/acp-2017-349/acp-2017-349-AC2-supplement.zip

---

## Author Response (AR1)

**Response to Anonymous Referee #1**

The manuscript reports flux data measured by the aerodynamic gradient (AGM) method over the canopy at two low-land forest sites in South China over a one-year period. Although the context of this study does not introduce new science beyond what has been known, such long-term flux datasets in forest ecosystem are rare and deserve consideration for publication. The manuscript is organized, read well and carefully drafted given the data that it presents. The data reporting sections (Section 3.1 – Section 3.4) are appropriately supported by data and the depth of conclusions/implication can benefit from incorporating the recent findings obtained from mass balance study and stable mercury isotope investigation in forest ecosystems. Overall, I recommend publication of the manuscript and have the following minor comments.

Thanks to the reviewers for giving us very useful comments to improve the manuscript. Detailed responses to reviewers' comment are list bellow.

Comments 1: The LAI and other descriptive characterization (e.g., other predominant vegetative species, forest floor coverage, radiation transfer under canopy) of the two forest sites should be provided for better assessment of the forest site.

Response 1: The LAI and other descriptive characterization (other predominant vegetative species, Canopy density, radiation transfer under canopy) of the two sites were added in table 1. And the characterization was used as an evidence for the source/sink discussion of Section 3.4

Comments 2: Appropriately determining the turbulent transfer coefficient (K) is critical for estimating AGM flux, yet it is not clear in the manuscript how K varies. It will be useful to report the estimated K values and its diurnal variation in different in different season for evaluation of the reasonableness of estimated K values.

Response 2: We agree the turbulent transfer coefficient (K) is critical for estimating GEM flux, and the diurnal variation in different season was provided in SI.

Comments 3: The quality control statement of the AGM flux measurement is limited only to the detection of Hg vapor but did not consider other sources of bias (the long tubing, flow synchronization, intermittent sampling at the two level, etc.) that might introduce uncertainty to the flux measurement. Would it be a possibility that these variables can also be assessed to better represent the measurement uncertainty?

Response 3: The blank experiments to measure the monitoring system error were conducted before the installation by placing the air intakes in the zero mercury gas (Zero Air Tank, Tekran Inc.) for 48h. There were almost no adsorption/emission from the monitoring system (including of the long Teflon tube, the soda-lime tank and the electromagnetic valves) with the measurement results less than the detection limit of the instrument (0.1 ng m-3). The manual calibration by placing the air intakes in certain Hg concentration (Tekran 2505, Tekran Inc.) for 24h were done once every one month. The recovery rates were between 95 to 105% with an average value of 100.3%.

Since the automatic GEM analyser switches between two gold tubes and gets a value every 5 min, the two concentrations were averaged in each 10 min (matched to the single height sampling interval by adjusting the time relay) to avoid possible bias caused by different gold tubes.

The 20min variations of GEM concentration at certain height were between -2% to 2% and -4% to 4% (95% confidence interval) at QYZ and HT sites respectively. Thus, the GEM concentration was in a semi-steady state during the sampling interval. The GEM concentration differences calculated as the average concentrations at the higher height minus the two adjacent average concentrations at the lower sampling height (all in 10 min interval) could reduce the residual error.

Comments 4: The characterization of the two sites (clean and contaminated) could cause confusion. Given the mean annual TGM concentrations (3.64 and 5.93 ng m-3),

both locations should have been considered under the influence of regional Hg emission plumes. I suggest using "mildly polluted" and "moderately polluted" to avoid the confusion.

Response 4: Revised

Comments 5: It would have been extremely useful if Hg flux was also measured over the forest floor under the canopy. Such data will help enhance the source/sink discussion of Section 3.4.

Response 5: The GEM fluxes were also measured over forest floor under the canopy at QYZ site, and the results (unpublished data) showed the soil manifest as net GEM sources at all the seasons ( $0.13 \pm 0.43$ ,  $1.54 \pm 1.78$ ,  $4.76 \pm 1.86$  and  $2.07 \pm 1.73$  ng m-2 h-1 in winter, spring, summer and fall) were added in SI, the discussion was added in Section 3.4.

Comments 6: Recent characterization of stable mercury isotopes in foliage of various ages and litter, along with the quantification of litterfall production and Hg deposition through litterfall all indicate that forest ecosystem is a net sink for Hg in remote regions. Such data are also supported by the vertical gradient measurement of TGM concentration from forest floor through well above forest canopy, showing increasing TGM concentration with sampling height. The primary conclusions of this study appear to contradict these recent findings, even given the elevated TGM concentration in air. Since the flux data above forest floor are not reported in the manuscript, it is not possible to determine if it is the re-emission of contaminated soil that leads to the overall Hg source term. An in-depth discussion regarding these discrepancy will significantly enhance the scientific value of the manuscript.

Response 6:There are ongoing debates regarding whether or not forest is a sink or a source of GEM because the forest/air exchange flux is the sum of vegetation and soil

exchange flux, depending on not only atmospheric concentration and meteorological conditions, but also plant community composition and soil structure (Bash and Miller, 2009; Converse et al., 2010) over shorter or longer periods

The study of foliage/atmosphere mercury exchange at QYZ indicated that the vegetation presented as a net GEM source with a value of 1.32 ng m-2 h-1 (2.19, 0.32 2.51, -0.01, 1.32 ng m-2 h-1 in winter, spring, summer and fall) caused by high rates of photoreduction and plant transpiration due to the high temperature and radiation, the larger leaf surface area and elevated mercury deposition (Luo et al., 2016). In addition, the study of the mercury exchange between atmosphere and soil under the forest canopy at QYZ through the DFC methods (unpublished data) showed the soil manifest as net GEM sources at all the seasons (0.13 ± 0.43, 1.54 ± 1.78, 4.76 ± 1.86 and 2.07 ± 1.73 ng m-2 h-1 in winter, spring, summer and fall), and the results were added in SI. Thus, the net emissions observed at QYZ were contributed by both soil and foliar emissions. The GEM fluxes over forest canopy (8.09 ng m-2 h-1) in this study were almost similar to the sum (7.27 ng m-2 h-1) of emission fluxes from foliage and soil in summer, but had lager values in other seasons. It might be because of the underestimation of the GEM fluxes from soil due to the decreased turbulence in chamber using the DFC method, and the lack of GEM fluxes from the undergrowth vegetation.

It is a pity that foliage/atmosphere and soil/atmosphere mercury exchange at HT have not been measured, respectively. However, the comparison of Hg content of currentyear foliage and soil between two sites might indicate that there were larger GEM emission fluxes from soil but much larger GEM adsorption by foliage.

In addition to the lager GEM concentration and soil Hg content, the in-depth discussion of discrepancy caused by many other factors including the flux quantification method, specific tree species (evergreen tree species generally have higher exchange capabilities of GEM relative to deciduous tree species), reemission from photoreduction and transpiration was added in Section 3.4.

**Responses to Anonymous Referee #2**

Thanks the reviewer for giving us very useful comments to improve the manuscript. Detailed responses to the comments are list bellow.

comments from Referees :Forest is an important ecosystem on the earth. Characterizing the role of forest in the global biogeochemistry cycling of Hg is an important research topic in global Hg cycling studies. This study investigated the GEM gradient at two typical forests in subtropical zone of China. GEM concentrations and GEM fluxes measured are valuable for the atmospheric Hg studies in regional and global scale. I don't see major errors in the sampling techniques and method, and the discussions, for most cases, are sound throughout the whole manuscript. I would suggest a publication after addressing the following questions.

Comments 1: Major points: The result from this study found the forest in subtropical zone of China is a net source of atmospheric GEM, and I have no doubt for this result. My question is, why the net emissions were observed in the study areas, soil emissions or foliar emissions? The authors did discuss this scientific issue in section 3.4, but we still don't know the exact causes. I think this is critical for atmospheric science. As we know, previous field observations on foliage/atmospheric Hg fluxes mostly revealed a net sink of GEM. If forest canopy is a net source in the study area, this would be an important finding. If the net emissions of GEM from forest were caused by elevated soil Hg emissions (soil Hg concentrations were elevated), then future studies regarding the mass balance of GEM in forest using litterfall approach approaches should also consider the soil emissions or reemissions. The net emissions observed in this study might be also due to many other factors including the specific tree species (evergreen tree species generally have higher uptake capabilities of GEM relative to deciduous tree species), lower leaf area index, reemission of dew water and transpiration stream, which should be also assessed in the manuscript.

Response 1: There are ongoing debates regarding whether or not forest is a sink or a

source of GEM because the forest/air exchange flux is the sum of vegetation and soil exchange flux, depending on not only atmospheric concentration and meteorological conditions, but also plant community composition and soil structure (Bash and Miller, 2009; Converse et al., 2010) over shorter or longer periods.

The study of foliage/atmosphere mercury exchange at QYZ indicated that the vegetation presented as a net GEM source, with a value of  $1.32 \text{ ng m}^{-2} \text{ h}^{-1}$  (2.19, 0.32 2.51, -0.01, 1.32 ng m-2 h-1 in winter, spring, summer, and fall, respectively), caused by high rates of photoreduction and plant transpiration due to high temperature and radiation, relatively large leaf surface area, and elevated mercury deposition (Luo et al., 2016). In addition, the study of the mercury exchange between atmosphere and soil under the forest canopy at QYZ through the DFC methods showed the soil also manifest as net GEM sources at all the seasons ( $0.13 \pm 0.43$ ,  $1.54 \pm 1.78$ ,  $4.76 \pm 1.86$  and  $2.07 \pm$ 1.73 ng m-2 h-1 in winter, spring, summer, and fall respectivley; unpublished data). The above results were added in SI. Thus, the net emissions observed at QYZ were contributed by both soil and foliar emissions. The GEM fluxes over forest canopy (8.09 ng m-2 h-1) in this study were almost similar to the sum (7.27 ng m-2 h-1) of emission fluxes from foliage and soil in summer, but had lager values in other seasons. It might be because of the underestimation of the GEM fluxes from soil due to the decreased turbulence in chamber using the DFC method, and the lack of GEM fluxes from the undergrowth vegetation. Although there were net GEM emissions (58.5  $\mu$ g m-2 yr-1) from forest in this study at QYZ site based on the measurement of the GEM fluxes over forest canopy, on account of extremely large Hg deposition (wet deposition: 14.4 µg m-  $^{2}$  yr-1; dry deposition: 52.5 µg m-2 yr-1; Luo et al., 2016), the forest presented as a Hg source, overall.

It is a pity that foliage/atmosphere and soil/atmosphere mercury exchange at HT have not been measured, respectively. However, the comparison of Hg content of currentyear foliage and soil between two sites might indicate that there were larger GEM emission fluxes from soil but much larger GEM adsorption by foliage.

In addition to the lager GEM concentration and soil Hg content, the in-depth discussion of discrepancy caused by many other factors including the flux quantification method,

specific tree species (evergreen tree species generally have higher exchange capabilities of GEM relative to deciduous tree species), reemission from photoreduction and transpiration was added in Section 3.4.

Comments 2: Minor points:

Line 79: why did the authors select the two sampling heights at QYZ and HT? Are you any pervious studies to support your setting; Will the chose of different sampling height affect the flux result? Line 81: 'half canopy height'? The lower sampling heights were 25 and 22.5 m agl, which were much higher than the canopy height.

Response 2:The result measured by AGM represent a mean value of GEM flux in regional area, i.e, footprints area of tower, which is related to the measuring height and meteorological conditions (Fritsche et al., 2008b). Previous study estimated that the footprint of intake at 40 m height on the flux tower was 100 - 400 m (Zhao et al., 2005). Therefore, the footprints of the intakes located at different height in our study may be similar due to the relatively uniform distribution of *pinus massoniana* or *cunninghamia lanceolata* forest within 500 m around the flux towers in our research.

Considering the extremely large disturbance of temperature and wind speed over forest canopy, especially close to the canopy, the lower air intake should be set at least half canopy height (Table 1) **above the canopy** to ensure the stability of the results (Lindberg et al., 1998). Thus, the lower sampling heights were about 1.5 times of the canopy height.

Comments 3: Section 3.1: I think the annual variations of GEM gradient at QYZ and HT are also important and should be presented in Figures. The authors should also show the annual, seasonal and diurnal trend of GEM gradient.

Response 3: The GEM gradient and the turbulent transfer coefficient (K) are both critical for estimating GEM flux, and the annual, seasonal and diurnal variation in different season were presented in SI.

Comments 4: Line 155: the authors should note the sampling height of the annual mean GEM concentration.

Response 4: The atmospheric GEM concentrations presented in manuscript were the average GEM concentrations at two heights, which was clarified in the revised manuscript.

Comments 5: Line 156: the global and northern hemisphere background should be referred to GMOS studies.

Response 5: A reference for GMOS studies was added.

Comments 6: Line 159: References are needed here

Response 6: Two references were added.

Comments 7: Line 161: Do you have any evidence for this hypothesis? I strongly suggest the authors analyze the source-receptor relationships at the sampling sites? Measurement of GEM in the atmosphere is also an negligible part of this study.

Response 7: The sentence was removed. The source-receptor relationships at QYZ site was analyzed by using the Hg content of precipitation and throughfall, and the fluxes of soil/atmosphere, foliage/atmosphere and forest/atmosphere, the discussion was added in Section 3.4.

Comments 8: Line 190: should be 'have positive values at QYZ'.

**Response 8: Revised**

Comments 9: Line 196: References are needed here.

Response 9: "See section 3.3" was added because there were many related references and discussion in section 3.3.

Comments 10: Line 227: the source from WS mercury mining area? Any evidence?

Response 10: The WS mercury mining area is located in the northwest of the HT site about 100 km away. The sudden rise of GEM concentration not only on May 14 presented in Figure 7, but also on January 17, September 5, October 10 and November 17 24 at HT site, corresponded to northwest wind prevailed according to the wind direction records. Thus, we believe that the sudden rise of GEM concentration might be caused by the approach of a high-mercury-content air mass from WS Mercury Mine leading by northwest wind.

Comments 11: Line 297: the study only reveal the whole forest is a net source, but not vegetation, you did not measure foliage/atmospheric Hg flux. Table-1: the relative abundance of major tree species should be listed.

Response 11:The study of foliage/atmosphere Hg fluxes was also conducted at QYZ site by using the DFC method. And the results were published in 2016, and could be used to explain the contribution of foliage to the whole forest GEM emission. The relative abundance of major tree species was listed in Table 1.

**List of all relevant changes made in the manuscript**

- 1. Using "mildly polluted" and "moderately polluted" to replace the "clean" and "contaminated", respectively.
- "The 20min variations of GEM concentration at certain height were between -2% to 2% and -4% to 4% (95% confidence interval) at QYZ and HT sites respectively.

Thus, the GEM concentration was in a semi-steady state during the sampling interval" and "The manual calibration by placing the air intakes in certain Hg concentration (Tekran 2505, Tekran Inc.) for 24h were done once every one month. The recovery rates were between 95 to 105% with an average value of 100.3%." were added in 2.3 and 2.4 session to assessed the measurement uncertainty

- 3. "The blank experiments to measure the monitoring system error were conducted before the installation by placing the air intakes in the zero mercury gas (Zero Air Tank, Tekran Inc.) for 48h. There were almost no adsorption/emission from the monitoring system (including of the long Teflon tube, the soda-lime tank and the electromagnetic valves) with the measurement results less than the detection limit of the instrument (0.1 ng m-3)." was added in the quality control session.
- "(the average concentration at two heights)" was added and ", because HT station was affected by WS Mercury Mine" was removed in the first paragraph in session 3.1.
- 5. "see section 3.3" was added in section 3.2 to support "the increase of solar radiation and air temperature would cause the increasing in GEM emission from soil and vegetation"
- 6. The sentence" the study of foliage/atmosphere mercury exchange at QYZ indicated that the vegetation presented as a net GEM source in all seasons" was reworded as "the study of foliage/atmosphere mercury exchange at QYZ indicated that the vegetation presented as a net GEM source as the total effects with a value of 1.32 ng m-2 h-1 (2.19, 0.32, 2.51 and -0.01 ng m-2 h-1 in winter, spring, summer and fall respectively) caused by high rates of photoreduction and plant transpiration due to high temperature and radiation, relatively large leaf surface area and elevated mercury deposition" in section 3.4.
- 7. "In addition, the study of the mercury exchange between atmosphere and soil under the forest canopy at QYZ using the DFC methods also showed the soil manifested as net GEM sources at all the seasons (Figure S6, 0.13 ± 0.43, 1.54 ± 1.78, 4.76 ± 1.86 and 2.07 ± 1.73 ng m-2 h-1 in winter, spring, summer and fall, respectively; unpublished data). Thus, the net emissions observed at QYZ were contributed by

both soil and foliar emissions. The GEM fluxes over forest canopy (8.09 ng m-2 h-1) in this study were almost similar to the sum (7.27 ng m-2 h-1) of emission fluxes from foliage and soil in summer, but had lager values in other seasons. It might be because of the underestimation of the GEM fluxes from soil due to the decreased turbulence in chamber using the DFC method, and the lack of GEM fluxes from the undergrowth vegetation. Although the foliage/atmosphere and soil/atmosphere mercury exchange at HT have not been measured, respectively, the comparison of Hg content of current-year foliage and soil between two sites might indicate that there were larger GEM emission fluxes from soil but much larger GEM adsorption by foliage." In section 3.4.

- 8. "with relatively higher LAI at all seasons", "Evergreen tree species generally have higher exchange capabilities of GEM relative to deciduous tree species and result in high rates of photoreduction and plant transpiration under the high temperature, solar radiation and soil Hg content." and "Although there were net GEM emissions (58.5 μg m-2 yr-1) from forest in this study at QYZ site based on the measurement of the GEM fluxes over forest canopy, on account of extremely large Hg deposition (wet deposition: 14.4 μg m-2 yr-1; dry deposition: 52.5 μg m-2 yr-1; Luo et al., 2016), the forest presented as a Hg source, overall." Were added in the last paragraph in session 3.4.
- 9. Two reference were added

**List of all relevant changes made in the Supplementary Information**

- 1. The catalogue was updated
- The figure "Annual variations of GEM gradient and turbulent transfer coefficient (K) at QYZ (a) and HT (b) stations." was added as Figure S1.
- The figure "Diurnal variations of GEM gradient and turbulent transfer coefficient (K) in each season." Was add as Figure S2
- 4. The figure "The diurnal variation of soil GEM emission fluxes, GEM concentrations and solar radiations in each seasons" was added as Figure S6.

5. The figure number was reword according to the new order.

[revised manuscript text omitted]

species (relative abundance) type | Pinus massoniana (86.5%)                     | Cunninghamia lanceolata (92.4%)   |  |  |
| Other predominant vagatative                                                 | Pinus elliottii; Quercus fabei; Vitex | Marsa japonica ; Ilex pirpurea;   |  |  |
| species                                                                      | negundo; Rhododendron plonch;                | Cyclosorus parasticus; Woodwardia |  |  |
| species                                                               | Ischaemum indicum                            | prolifera                         |  |  |
| Forest age                                                                   | 31                                           | 27                                       |  |  |
| Canopy height (m)                                                            | 16                                           | 14                                       |  |  |
| Leaf area index (LAI) in
summer                             | 4.31                                  | 7.00                              |  |  |
| Canopy density                                                               | 0.7                                   | 0.8                               |  |  |
| Radiation transfer under canopy                                              | 3.0%                                  | 2.7%                              |  |  |
| Dominant soil type (Chinese soil name)                                       | Udic Ferrisols (Red Earth)                   | Haplic Acrisol (Yellow Earth)            |  |  |
| Organic matter content in surface soil (g kg -1 ) a    | 10~15                                        | 28.3                                     |  |  |
| Soil pH a                                                         | 4.52                                         | 3.85                                     |  |  |
| Annual average GEM concentration (ng m -3 ) b          | $3.64 \pm 1.82$                              | $5.93 \pm 3.16$                          |  |  |

| Hg content in soil organic layer
(ng g -1 ) c    | $76.2 \pm 6.0$ | 153 ± 28 |
|---------------------------------------------------------------------------|----------------|----------|
| Hg content in surface (0~5 cm)
soil (ng g -1 ) c | 42.6 ± 2.3     | 167 ± 32 |

587 a Data of QYZ and HT stations according to Gao et al. (2014) and Wang et al. (2009), respectively;

588 b Mean value of the measurements at the height of 25 m and 35 m at QYZ site, 22.5 and 30.5 m at HT site;

589 590 591 c Analyzed based on 18 samples using a direct Hg analyzer (DMA80, Milestone Inc., Italy).

Table 2. Pearson's correlation coefficient between GEM flux over forest canopy and atmospheric GEM concentration or each environmental

factor.

| Factors           | Sites | Winter   | Spring   | Summer   | Fall     |
|-------------------|-------|----------|----------|----------|----------|
| GEM concentration | QYZ   | -0.142** | -0.155** | 0.014    | -0.141** |
| GEW concentration | HT    | -0.232** | -0.226** | -0.197** | -0.183** |
| Air temperature   | QYZ   | 0.272**  | 0.166**  | 0.31**   | 0.298**  |
| An emperature     | HT    | 0.143**  | 0.121**  | 0.188**  | 0.135**  |
| Air humidity      | QYZ   | -0.314** | -0.003   | -0.293** | -0.339** |
| An numberry       | HT    | -0.101*  | -0.149** | -0.246** | -0.255** |
| Wind speed        | QYZ   | 0.159**  | 0.176**  | 0.162**  | 0.166**  |
| while speed       | HT    | 0.119**  | 0.180**  | 0.106**  | 0.162**  |
| Soil temperature  | QYZ   | 0.025    | 0.165**  | 0.288**  | 0.175**  |
| Son temperature   | HT    | 0.015    | 0.174**  | 0.253**  | 0.201**  |
| soil moisture     | QYZ   | 0.102**  | -0.198** | 0.03     | -0.106** |
| son moisture      | HT    | 0.001    | -0.032   | -0.003   | 0.034    |
| Radiation         | QYZ   | 0.628**  | 0.403**  | 0.401**  | 0.209**  |
| Radiation         | HT    | 0.265**  | 0.212**  | 0.313**  | 0.201**  |

596 597 \* Significant at p < 0.01 level; \*\* Significant at p < 0.001 level.

| Vegetation type        | Location                            | winter | spring | summer | fall      | GEM
con | method | Data source                            |
|------------------------|-------------------------------------|--------|--------|--------|-----------|------------|--------|----------------------------------------|
| Subtropical coniferous | Jiangxi province,
China          | 5.49   | 5.25   | 8.09   | 7.86      | 3.64       | AGM    | QYZ site                               |
| forest                 | Hunan province,
China            | -3.62  | 0.83   | 4.40   | -
0.40 | 5.93       | AGM    | HT site                                |
| Mature hardwood        |                                     | _      | _      | 10-330 | _         | 2.23       | MBR    |                                        |
| Yang pine plantation   | Tennessee, USA                      | _      | _      | _      | 17-
86 | 1.45       | MBR    | Lindberg et
al. (1998) a |
| Boreal forest          | Lake Gardsjon,
Sweden            | _      | _      | 1-4    | _         | 2.02       | MBR    | × ,                                    |
| Deciduous forest       | Connecticut, USA                    | _      | _      | 21.9   | _         | 1.34       | REA    | Bash and
Miller (2008)
b         |
| Deciduous forest       | Coventry
Connecticut,
England | _      | _      | -1.54  | _         | 1.41       | REA    | Bash and
Miller (2009)              |
| Meadow                 | Fruebuel,                           | 4.1    | -4.8   | 2.5    | 0.3       | 1.29       | AGM    | Converse et                            |
| Weadow                 | central
Switzerland              | -2.9   | -1.5   | 3.2    | -3.0      | 1.29       | MBR    | al. (2010)                             |

Table 3. Comparison of the GEM flux  $(ng \cdot m^{-2} \cdot h^{-1})$  from different the observations.

 a mean value (90% confidence interval), only measured during daytime;
 b median value of TGM (total gaseous mercury) flux